# PROOF OF FORGEABILITY: UNIVERSAL REPUDIATION AGAINST MEMBERSHIP INFERENCE ATTACKS

## ABSTRACT

Membership inference attacks (MIAs) aim to infer whether a data point was used to train a target model and are widely used to audit the privacy of machine learning (ML) models. In this work, we present a new approach to asserting repudiation evidence against MIA-supported claims. Existing strategies require computationally intensive, case-by-case proofs. We introduce Proof of Forgeability (PoF), which denies all membership claims with an universal repudiation. The key idea is to generate forged examples that are non-members yet are misclassified as members by MIAs. We construct forged examples by adding carefully designed perturbations to non-members so that the attack signal distribution derived from model outputs for the forged examples matches that of members. To achieve this, we use quantile matching to derive a member-like signal estimator (MLSE) that maps each non-member's signal to its target member-like signal. We prove the optimality of this MLSE and derive closed-form expressions when the attack signal is the logit-scaled true-label confidence. We then apply a first-order Taylor expansion of the signal with respect to the input to bridge the input and signal space. This relation converts the target signal change into an input perturbation and yields the designed perturbation in closed form. Empirical results demonstrate that the forged examples indeed confuse the MIAs in comparison with the genuine members; meanwhile, the forged examples differ imperceptibly from the original non-members in input content while fully preserving data utility.

## 1 INTRODUCTION

Machine learning (ML) models now proliferate across critical domains, including finance (Hernandez Aros et al., 2024) and healthcare (Zhang et al., 2022). However, modern ML models are vulnerable to leakage of sensitive training data (Papernot et al., 2016). Membership inference attacks (MIAs) (Shokri et al., 2017) are currently the most widely employed approaches for auditing the privacy of ML models. Government agencies, including the UK Information Commissioner's Office (ICO) and the US National Institute of Standards and Technology (NIST), have highlighted MIAs as a potential violation of confidentiality and a privacy threat to training data (Murakonda and Shokri, 2020). MIAs aim to infer whether a specific data point was included in the training dataset of a target model. A data point that was included is a *member*, and one that was not is a *non-member*. The inferred *training data membership* supports audits of privacy risk, assessments of copyright compliance, and broader AI safety evaluations (Liu et al., 2025). Numerous studies have advanced MIAs' performance and demonstrated their practical utility for auditing training data leakage across diverse models (Carlini et al., 2022; Zarifzadeh et al., 2024).

There is growing demand for verifiable evidence to substantiate claims of privacy leakage and copyright infringement, driven by AI regulations that continue to evolve and become more clearly defined (Voigt and Von dem Bussche, 2017). In response, the reliability of dominant MIA methodologies is receiving increased attention. To illustrate, suppose an *adversary* performs MIA on a query $(\mathbf{x}_q, \mathbf{y}_q)$ and infers training data membership. Is this evidence sufficient to conclude that $(\mathbf{x}_q, \mathbf{y}_q)$ was used to train the target model? Following Chowdhury et al. (2025), we pose the central question:

*Can a model owner plausibly deny a membership inference claim in practice?*

Figure 1: Court analogy comparing two repudiation strategies against a plaintiff's copyright claim supported by MIAs applied to a query. Proof-of-Repudiation (PoR): the defendant provides a detailed training log showing that the target model can be obtained from a dataset without the queried sample; this process must be repeated for each queried claim. Proof of Forgeability (PoF, ours): the defendant shows that any non-member can be perturbed imperceptibly to produce a forged example that MIAs misclassify as a member. Unlike PoR, PoF serves as a once-for-all repudiation across queries.

An explicit repudiation would require disclosing the entire training dataset, which is infeasible in practice. An implicit approach is to present a *Proof-of-Repudiation* (PoR), which demonstrates that the target model is reproducible from an alternative dataset that excludes the query $(\mathbf{x}_q, \mathbf{y}_q)$. A verifiable PoR supports the counterclaim that $(\mathbf{x}_q, \mathbf{y}_q)$ is *de facto* a non-member. This undermines the MIA inference and may deter the adversary from pursuing legal action. While a PoR suffices to raise reasonable doubt for a single claim, producing PoRs for every query incurs substantial computational overhead. These computational burdens motivate a further question:

*Can a model owner plausibly deny all membership inference claims with an universal repudiation?*

We propose to demonstrate that membership inference claims are forgeable by constructing non-members that MIAs would infer as members. We call this evidence *Proof-of-Forgeability* (PoF) and refer to the constructed non-members as *forged examples*. A PoF discredits the membership evidence by demonstrating that it can be replicated with *forged examples*, thereby enabling a universal repudiation of all membership claims. The comparison of these two repudiations in a court analogy is illustrated in Fig. 1.

To make PoF convincing, forged examples must satisfy three conditions: (1) they are excluded from the target model's training set (i.e., non-members); (2) state-of-the-art (SOTA) MIAs typically infer them as members; and (3) they remain within the underlying data distribution of training data. We achieve this forgeability objective under the specified conditions by *adversarial example generation*. Specifically, we add carefully crafted perturbations to non-members so that SOTA MIAs *cannot* distinguish them from members. Notably, MIAs typically distinguish members from non-members based on the *attack signal* derived from target model outputs (Zhu et al., 2025). A common signal is the true label confidence (TLC), the predicted probability assigned to the ground-truth label (Carlini et al., 2022). Different MIAs apply distinct scoring functions and decision rules over such signals to classify a query as a member or a non-member (Zarifzadeh et al., 2024). Therefore, it suffices to fool these MIAs by matching the signal distribution of forged examples to that of members.

Matching signal distributions presents two challenges. First, under the assumption that members and non-members are sampled from the same underlying data distribution, there exists a per-example correspondence between the signal an example would produce as a member and as a non-member. Estimating this correspondence reduces to deriving, for each non-member, its corresponding member-like signal, which we call the member-like signal estimator (MLSE). We derive the MLSE using a quantile matching transformation and prove its optimality under this task. For the logit-scaled TLC, which is approximately Gaussian (Carlini et al., 2022), we further derive closed-form expressions for the MLSE. Second, the distribution matching is realized in signal space, whereas the perturbation used to construct forged examples operates in data space. To bridge this gap, we employ a first-order Taylor expansion of the signal with respect to the input to relate input perturbations to the induced changes in signal space. This relation yields closed-form perturbation

magnitudes based on the discrepancy between member-like and original signal for each non-member query.

Empirically, we conduct extensive experiments to validate the properties of the forged examples across standard datasets, including CIFAR-10/100 and CINIC-10. First, we show that it is computationally feasible to generate such forged examples with indistinguishability, such that SOTA MIAs cannot distinguish them from the genuine members. Second, we find that these forged examples are imperceptible relative to the original non-members, as evidenced by minor input changes and comparable data utility. These findings motivate a reassessment of how current MIAs quantify privacy leakage in ML models and the development of robust MIAs that remain effective against forged examples.

In summary, our contributions are summarized as follows:

- We introduce Proof of Forgeability (PoF), a single repudiation mechanism that applies to all membership inference claims (§3.1).

- We propose an algorithm for generating forged examples that is backed by rigorous theoretical analysis and derivation. This elucidates why forged examples induce member-like attack signals and thus evade MIAs (§3.2 and §3.3).

- Extensive experiments across datasets, MIAs, and attack configurations demonstrate that the forged examples successfully evade SOTA MIAs, while differing imperceptibly from the original non-members in both input contents and data utility (§4.2).

## 2 RELATED WORKS

### 2.1 MEMBERSHIP INFERENCE ATTACKS (MIAS)

MIAs (Shokri et al., 2017) aim to predict whether a data point was included in the training set of a target model. Adversaries typically rely on the target model's outputs as *attack signal* to classify a query example as a member or a non-member (Zhu et al., 2025). Specifically, the attacker compares the query's signal against the distributions of member and non-member signals and then decides membership accordingly (Carlini et al., 2022). Numerous studies have enhanced this framework by extracting more fine-grained information to characterize these two distributions. For instance, Ye et al. (2022) trains multiple reference models to simulate the signal distributions empirically. LiRA (Carlini et al., 2022) formalizes this framework as a likelihood ratio test and employs a parametric method to estimate the signal distributions. Building on these advances, Zarifzadeh et al. (2024) leverages both population data and reference models to improve attack power and robustness. Despite their remarkable performance, these methods primarily model and compare attack-signal distributions. This raises a question: if one constructs a forged non-member dataset whose attack signals match those of members, would SOTA MIAs fail to provide reliable privacy auditing? In this work, we illustrate how to generate such data using adversarial example generation to evade MIAs.

### 2.2 FORGEABILITY AND PROOF-OF-REPUDIATION

Forgeability (Thudi et al., 2022) was introduced in the context of machine unlearning (Bourtoule et al., 2021). Informally, two datasets are forgeable if training on either dataset obtains the same final weights, up to a small error. This obtainity is certified by a *Proof-of-Learning* (PoL) log (Jia et al., 2021), which records the training trajectory from initialization to the final weights, including the sequence of data points. The training rule refers to an update operator $g$ that maps a checkpoint and the data used at that step to the next checkpoint. Given a PoL log, one can verify its validity by reproducing its computation. Specifically, a verifier reproduces the checkpoint at $t$ using the items in the log, including the $(t-1)$-th checkpoint, data points used at step $t$, and the same update rule $g$. The verifier then computes the distance between the logged $t$-th checkpoint and the reproduced one in the parameter space. This distance is called the verification error, and the update at step $t$ is acceptable if the verification error is below a prescribed threshold.

Proof-of-Repudiation (PoR) (Kong et al., 2023) is a special case of PoL that empowers the model owner to repudiate the membership claim. Given a target model and a claim that data point $x^*$ is a member of its training dataset $D$, a valid PoR is essentially the PoL log that records a training

trajectory obtaining the same model from an alternative dataset $D^-$ that excludes $x^*$. This provides verifiable evidence that the model could have been obtained without using $x^*$ (Kong et al., 2022). Although a PoR log can repudiate the claim, generating and verifying such logs can be computationally expensive. This motivates more efficient repudiation mechanisms.

## 2.3 ADVERSARIAL EXAMPLES

Adversarial examples (Szegedy et al., 2013) are minimally perturbed inputs that induce ML models to misclassify while they retain high accuracy on unperturbed data. One widely-used and efficient method to generate such examples is the fast gradient sign method (FGSM) (Goodfellow et al., 2014), which perturbs the input along the element-wise sign of the loss gradient under an $L_\infty$ constraint. Let $J(\cdot)$ denote the loss and let $\boldsymbol{\theta}$ be the model weights. For an input-label pair $(\mathbf{x}, \mathbf{y})$, define the gradient with respect to the input as $\mathbf{g} = \nabla_{\mathbf{x}}(J(\boldsymbol{\theta}, \mathbf{x}, \mathbf{y}))$. The FGSM adversarial example is

$$\mathbf{x}^{adv} = \mathbf{x} + \epsilon \cdot \mathrm{sign}(\mathbf{g}), \tag{1}$$

where $\mathrm{sign}(\cdot)$ is applied element-wise and $\epsilon$ is the $L_\infty$ perturbation budget. Kurakin et al. (2016) refined FGSM to an iterative variant, I-FGSM, which improves attack success under the same budget and enforces valid input bounds. With step size $\alpha$ and $T$ iterations, the update rule are

$$\mathbf{x}_{t+1}^{\mathrm{adv}} = \mathrm{Proj}_{\mathbf{x}}^{\epsilon}\left(\mathrm{clip}_{\mathcal{X}}\left(\mathbf{x}_t^{\mathrm{adv}} + \alpha \cdot \mathrm{sign}\left(\nabla_{\mathbf{x}} J(\boldsymbol{\theta}, \mathbf{x}_t^{adv}, \mathbf{y})\right)\right)\right), \text{ for } i = 1, \ldots, T-1, \tag{2}$$

where $\mathbf{x}_0^{\mathrm{adv}} = \mathbf{x}$, $\mathrm{Proj}_{\mathbf{x}}^{\epsilon}$ projects onto the $L_\infty$ ball of radius $\epsilon$ centered at the original input $\mathbf{x}$, and $\mathrm{clip}_{\mathcal{X}}$ enforces the valid input domain. A common choice of $\alpha$ is $\epsilon/T$. Below, we follow this paradigm to generate forged examples, and the main challenge here is to estimate an appropriate perturbation budget of $\epsilon$. After that, Later works extended this line of research in two main directions: optimization-based attacks (Lin et al., 2020; Dong et al., 2018) and augmentation-based attacks (Xie et al., 2019; Yun et al., 2024; Lin et al., 2020; Li et al., 2023; Wang et al., 2021).

# 3 CONSTRUCTION OF PROOF-OF-FORGEABILITY

## 3.1 PROBLEM FORMULATION

Formally, let the target classifier be $f_{\boldsymbol{\theta}_t} : \mathcal{X} \to \Delta^{K-1}$, parametrized by $\boldsymbol{\theta}_t$, and trained on dataset $D_M = \{(\mathbf{x}_i^M, \mathbf{y}_i^M)\}_{i=1}^{n^M}$. The dataset $D_M$ consists of independent and identically distributed (i.i.d.) samples drawn from the underlying data distribution $P_{data}$ over $\mathcal{X} \times \mathcal{Y}$. Here, $\mathcal{X}$ denotes the input space and $\Delta^{K-1}$ is the probability simplex over $K$ classes. Each ground-truth $\mathbf{y}_i \in \{0, 1\}^K$ is represented as a one-hot vector. We also define a non-member dataset $D_N = (\mathbf{x}_i^N, \mathbf{y}_i^N)_{i=1}^{n^N}$ consists of i.i.d. samples from the same distribution $P_{data}$, with $D_N \cap D_M = \emptyset$ to ensure that non-members are excluded from training set of $f_{\boldsymbol{\theta}}$. Define an attack signal function $s : \Delta^{K-1} \times \{1, \ldots, K\} \to \mathcal{S}$, that maps the target model's predicted probability vector and the ground-truth label to a signal in the signal space $\mathcal{S}$ used for membership inference. We consider $s$ to be scalar-valued, since SOTA MIAs typically adopt scalar signals such as LOSS or the TLC and its variants. Suppose the underlying distribution of the attack signal over members and non-members is $S^M$ and $S^N$, respectively. Let $S^M$ and $S^N$ denote the distribution of the attack signals for members and non-members. Using $s$, we form empirical signal samples for $S^M$ and $S^N$ as $\{s(f(\mathbf{x}_i^M), \mathbf{y}_i^M)\}_{i=1}^{n^M} = \{s_i^N\}_{i=1}^{n^M}$ and $\{s(f(\mathbf{x}_i^N), \mathbf{y}_i^N)\}_{i=1}^{n^N} = \{s_i^N\}_{i=1}^{n^N}$. These empirical samples approximate draws from $S^M$ and $S^N$.

PoF seeks to construct forged examples $\{\mathbf{x}_i^F, \mathbf{y}_i^F\}_{i=1}^{n^M}$ from non-member data $\{\mathbf{x}_i^N, \mathbf{y}_i^N\}_{i=1}^{n^M}$, using the adversarial example generation framework, such that the empirical distribution of the resulting attack signals matches that of members. Let $S^F$ denote the distribution of the attack signal for forged examples. The empirical samples for $S^F$ are $\{s(f_{\boldsymbol{\theta}}(x_i^F), y_i^F)\}_{i=1}^{n^N} = \{\mathbf{s}_i^F\}_{i=1}^{n^N}$. PoF aims to make $S^F$ indistinguishable from $S^M$, the distribution of member signals, thereby inducing MIAs to misclassify forged examples as members. In this work, we illustrate how to generate such forged examples for scalar-valued signal functions. The overall pipeline is illustrated in Fig. 2.

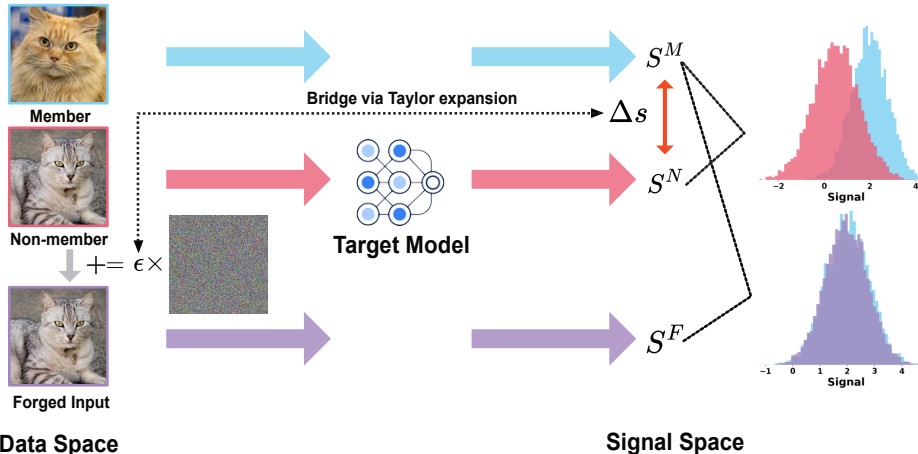

Figure 2: Overview of the Proof-of-Forgeability pipeline. Members and non-members enter the target model to obtain their signals $S^M$ and $S^N$, respectively. We omit the labels here for the ease of presentation. A non-member is perturbed by a small, bounded change to create a forged example. Taylor expansion bridges data space and signal space and yields a proper perturbation that shifts $S^N$ toward its estimated member-like counterpart. The signal distribution of forged examples $S^F$ matches that of members, which fools MIAs operating on output signals to infer membership.

## 3.2 MEMBER-LIKE SIGNAL ESTIMATOR

Carlini et al. (2022) formulated membership inference as a likelihood-ratio test (LRT). Given a query example $(\mathbf{x}_q, \mathbf{y}_q)$ and a target model $f_{\boldsymbol{\theta}_t}$, the LRT statistics is

$$\Lambda(f_{\boldsymbol{\theta}_t}; \mathbf{x}_q, \mathbf{y}_q) = \frac{p\big(\boldsymbol{\theta}_t \mid \mathbb{Q}_{\text{in}}(\mathbf{x}_q, \mathbf{y}_q)\big)}{p\big(\boldsymbol{\theta}_t \mid \mathbb{Q}_{\text{out}}(\mathbf{x}_q, \mathbf{y}_q)\big)}, \tag{3}$$

where for $b \in \{\text{in}, \text{out}\}$, $p\big(\boldsymbol{\theta}_t \mid \mathbb{Q}_b(\mathbf{x}_q, \mathbf{y}_q)\big)$ denotes the probability density of $\boldsymbol{\theta}_t$ under the model-parameter distribution $\mathbb{Q}_b(\mathbf{x}_q, \mathbf{y}_q)$. For the remainder of the paper, we write $\mathbb{Q}_b$ as shorthand for $\mathbb{Q}_b(\mathbf{x}_q, \mathbf{y}_q)$. Here, $\mathbb{Q}_{\text{in}}$ and $\mathbb{Q}_{\text{out}}$ are the distributions over model parameters induced by training on datasets that include, or exclude, the query example, respectively. Since $\mathbb{Q}_{\text{in}}$ and $\mathbb{Q}_{\text{out}}$ are analytically intractable and often inaccessible in the black-box setting, recent works typically employ low-dimensional surrogates, $\tilde{\mathbb{Q}}_{\text{in}}$ and $\tilde{\mathbb{Q}}_{\text{out}}$, that serve as proxies for the intractable parameter distributions. These surrogates are defined as distributions of attack signals computed at $(\mathbf{x}_q, \mathbf{y}_q)$ across models trained with, or without, the query example. The selected attack signal should correlate with the underlying parameters while remaining efficient to compute. Common choices include the loss value (Shokri et al., 2017) and variants of the TLC (Carlini et al., 2022; Zarifzadeh et al., 2024).

For each query $(\mathbf{x}_q, \mathbf{y}_q)$, the adversary compute the attack signal $s_q = s(f_{\boldsymbol{\theta}}(\mathbf{x}_q), \mathbf{y}_q)$, and compare the likelihoods $p(s_q|\tilde{\mathbb{Q}}_{\text{in}})$ and $p(s_q|\tilde{\mathbb{Q}}_{\text{out}})$. Take TLC as an example, $p(s \mid \tilde{\mathbb{Q}}_{\text{in}})$ assigns high probability density on larger values than $p(s \mid \tilde{\mathbb{Q}}_{\text{out}})$ due to model overconfidence (Chen and Pattabiraman, 2023). Consequently, a non-member query tends to produce a smaller TLC signal and therefore has a higher likelihood under $\tilde{\mathbb{Q}}_{\text{out}}$. Our goal is to transform each such signal induced by a non-member query that is more consistent with $\tilde{\mathbb{Q}}_{\text{in}}$, namely, a value whose likelihood under $\tilde{\mathbb{Q}}_{\text{in}}$ exceeds its likelihood under $\tilde{\mathbb{Q}}_{\text{out}}$. This amounts to estimating, for each non-member query, the *member-like signal* that the query would have produced as if it had been included in training. To this end, we propose a unified *member-like signal estimator* (MLSE) that maps a non-member's signal to its corresponding member-like signal.

Note that both the member and non-member data of the target model are sampled from the same underlying data distribution $P_{\text{data}}$. Hence, the discrepancy between the signal distributions of members and non-members is attributable to the training process. This observation implies that any alignment should correct the training-induced shift rather than a data-distribution mismatch. We therefore posit that the optimal MLSE should align the distribution of the member-like signals with that of true member signals at minimal transport cost. To achieve this, we employ a *quantile match-*

*ing transformation*, where each non-member signal is paired with the member signal that shares the same percentile rank in its respective empirical distribution. Formally, let $F_N$ and $F_M$ represent the cumulative distribution functions (CDFs) of the non-member and member signal distributions, $S^M$ and $S^N$, respectively. For a non-member signal $s_i^N$, the forged member-like signal is

$$s_i^F = F_M^{-1}(F_N(s_i^N)), \tag{4}$$

where $F_M^{-1}$ is the quantile function, i.e., the generalized inverse CDF. This establishes a correspondence between $S^N$ and $S^M$ that aligns their CDFs. Quantile matching is monotonic and avoids density estimation. As demonstrated in Lemma 3.1, this correspondence is the optimal transport solution in one dimension for any convex cost on the displacement. Therefore, the quantile matching transformation is theoretically optimal for the MLSE when signals are scalar-valued under common distance metrics. It is also stable and parameter-free. The proof of Lemma 3.1 is provided in App. B.2.

**Lemma 3.1** (Quantile matching is one-dimensional optimal transport). *Consider two atomless probability measures $\mu$ and $\nu$ on $\mathbb{R}$ with strictly increasing CDFs $F_\mu$ and $F_\nu$, and quantile functions $Q_\mu = F_\mu^{-1}$, $Q_\nu = F_\nu^{-1}$. The quantile matching map is $T(s) = Q_\nu(F_\mu(s)) = F_\nu^{-1} \circ F_\mu(s)$.*

*Then the map $T$ is the unique optimal transport map that minimizes the expected cost $\mathbb{E}_{s \sim \mu}[c(s, T(s))]$ for any cost function $c(s_1, s_2) = h(s_2 - s_1)$, where $h : \mathbb{R} \to \mathbb{R}$ is strictly convex.*

Furthermore, we can reduce the computational cost of empirical estimation for certain attack signals. When using parametric methods for modeling $\mathbb{Q}_{\text{in}}$ and $\mathbb{Q}_{\text{out}}$ with fewer reference models, Carlini et al. (2022) adopts logit-scaled TLC, which empirically follows a normal distribution. For such signals with a parametric form, we can derive a closed-form expression for the MLSE. Theorem 3.2 establishes this result for logit-scaled TLC, and the proof is provided in App. B.3.

**Theorem 3.2** (Closed-form for Gaussian-distributed Signal). *Assume the one-dimensional signals for members follow $S^M \sim \mathcal{N}(\mu_M, \sigma_M^2)$ and for non-members follow $S^N \sim \mathcal{N}(\mu_N, \sigma_N^2)$. Then for a non-member signal $s_i^N$, its member-like signal $s_i^F$ via quantile matching is*

$$s_i^F = \mu_M + \frac{\sigma_M}{\sigma_N}(s_i^N - \mu_N). \tag{5}$$

*The target signal change $\Delta s_i$ is*

$$\Delta s_i = s_i^F - s_i^N = (\mu_M - \mu_N) + \left(\frac{\sigma_M}{\sigma_N} - 1\right)(s_i^N - \mu_N). \tag{6}$$

### 3.3 Bridge Input and Signal Space via Taylor Expansion

In the previous subsection, we justified the fraud to MIAs at the signal level and proposed estimating, for each non-member signal, a corresponding member-like signal. Building on this idea, we now construct forged examples from these estimated member-like signals. To substantiate the PoF, these forged examples must satisfy three conditions: (1) they are excluded from the target model's training set and are therefore non-members, (2) SOTA MIAs typically infer them as members, and (3) they remain within the underlying data distribution $P_{\text{data}}$. Based on the analysis in §3.2, condition (2) holds if the signal distribution of the forged examples matches that of the members. Moreover, conditions (1) and (3) are satisfied when forged examples are produced by adding *imperceptible* perturbations to non-members. This construction preserves the non-membership of forged examples with respect to the fixed training set and keeps them within the support of the underlying data distribution $P_{\text{data}}$.

We instantiate the imperceptible perturbation using adversarial example generation methods. These methods add carefully designed perturbations to inputs along the steepest ascent direction of the loss function to induce model misclassifications. In this context, for a non-member query $(\mathbf{x}_q^N, \mathbf{y}_q^N)$, the forged input has the form of

$$\mathbf{x}_q^F = \mathbf{x}_q + \epsilon_q \cdot \text{sign}(\mathbf{g}_q), \tag{7}$$

where $\epsilon_q \geq 0$ controls the perturbation magnitude, and $\mathbf{g}_q = \nabla_{\mathbf{x}}\big(s(f_{\boldsymbol{\theta}_t}(\mathbf{x}_q), \mathbf{y}_q)\big)$ is the gradient of the attack signal with respect to the input, evaluated at the target model $f_{\boldsymbol{\theta}_t}$. The sign operator is applied element-wise. The forged input, paired with the original non-member label, forms the

forged example $(\mathbf{x}_q^F, \mathbf{y}_q^N)$. For brevity, we present a single-step update and omit iterative refinements and projection to the valid input range. To satisfy condition (2), the signal distribution of the forged examples across all non-member queries must align with the members' signal distribution. As established in §3.2, this distribution is identical to the distribution of the member-like signals for non-members. Let $s_q^N$ and $s_q^F$ denote the original signal and the member-like signal for a non-member query $(\mathbf{x}_q, \mathbf{y}_q)$. We therefore choose an appropriate perturbation magnitude $\epsilon_q$ such that the forged example's signal equals its member-like signal, namely $s\big(f_{\boldsymbol{\theta}_t}(\mathbf{x}_q^F), \mathbf{y}_q^N\big) = s_q^F$. This choice aligns the signal distribution of forged examples with that of the members and thereby satisfies condition (2).

While the move from the non-member signal $s_q^N$ to the member-like signal $s_q^F$ is defined in signal space, the perturbation used to construct forged examples operates in input space. We therefore require a bridge to link the desired signal change to the perturbation magnitude in input space. As stated in Lemma 3.3, a first-order Taylor expansion of the signal with respect to the input provides this bridge by relating small input perturbations to the induced change in the signal. This relation yields a closed-form expression for the perturbation magnitude as a function of the discrepancy between the member-like and original signal for each non-member query. The proof of Lemma 3.3 is provided in App. B.1.

**Lemma 3.3** (Bridge Input and Signal Space via Taylor Expansion). *Let $s$ be a differentiable scalar attack signal. For each non-member query $(\mathbf{x}_q, \mathbf{y}_q)$ with signal $s_q^N$, let $s_q^M$ denote its member-like signal, and define the target signal change $\Delta s_q = s_q^M - s_q^N$. Let $\mathbf{g}_q = \nabla_{\mathbf{x}} s\big(f_{\boldsymbol{\theta}_t}(\mathbf{x}_q^F), \mathbf{y}_q^N\big)$ be the gradient of the attack signal with respect to the input. Assume a perturbation $\delta_{\mathbf{x}} = \epsilon \cdot \mathrm{sign}(\mathbf{g})$ along this gradient sign direction, where $\mathrm{sign}(\cdot)$ denotes the element-wise sign function and $\epsilon_q > 0$ is the perturbation magnitude. Under the first-order Taylor approximation, the closed-form for $\epsilon_q$ is:*

$$\epsilon_q = \frac{\Delta s_q}{\|\mathbf{g}_q\|_1}, \tag{8}$$

*where $\|\cdot\|_1$ denotes the $l_1$ norm.*

In summary, we demonstrate how to construct eligible forged examples to constitute convincing Proof-of-Forgeability. Specifically, the model owner first samples non-members from the data distribution $P_{\mathrm{data}}$, and estimates their member-like signals using MLSE. For each non-member query, the discrepancy between the member-like and original signal determines an appropriate perturbation magnitude. In implementation, we apply the I-FGSM Kurakin et al. (2016) with these magnitudes to the non-members, producing forged examples that typically evade SOTA MIAs. The pseudo-codes of PoF are shown in Alg. 1.

## 4 EXPERIMENTS

We conduct experiments to validate the following properties of the constructed forged examples.

**Indistinguishability** Forged examples cannot be distinguished from genuine members by SOTA MIAs across data augmentation settings and across different numbers of reference models.

**Imperceptibility** Forged examples differ minimally from their corresponding original non-member data, preserving the input contents and data utility.

### 4.1 EXPERIMENTAL SETUP

**Datasets and implementations.** We evaluate our methodology on three publicly accessible benchmarks: CIFAR-10, CIFAR-100 (Krizhevsky et al., 2009), and CINIC-10 (Darlow et al., 2018).

For a fair comparison, we employ Wide-ResNet (Zagoruyko and Komodakis, 2016) as the backbone across all datasets and adopt an identical training protocol following established conventions (Carlini et al., 2022; Zarifzadeh et al., 2024). This protocol fixes the optimizer, learning rate schedule, data augmentation, and regularization to match the baseline configuration. Across all datasets, we follow Carlini et al. (2022) to **randomly partition the training set into two disjoint, equal-sized halves**. One-half is used to train the target model, and these examples are treated as *members*. The other half is held out strictly for evaluation as *non-members*. This creates a 50/50 member versus non-member split drawn from the same underlying distribution.

Table 1: Comparison of MIA performances on forged examples constructed using different guiding signals. We report AUC and TPR at FPRs of 0.01% and 0.0%. A lower TPR at low FPR indicates stronger indistinguishability, and an AUC near 50% corresponds to chance and thus indicates successful forgeability. N/A* denotes evaluation on the original non-members. The LiRA- and RMIA-based guiding signals are variants of TLC and are detailed in App. C.3.

| Guiding Signal | Attack | CIFAR-10 | | | CIFAR-100 | | | CINIC-10 | | |
|---|---|---|---|---|---|---|---|---|---|---|
| | | AUC | TPR@FPR | | AUC | TPR@FPR | | AUC | TPR@FPR | |
| | | | 0.01% | 0.0% | | 0.01% | 0.0% | | 0.01% | 0.0% |
| N/A* | Attack-R | 64.63 | 1.84 | 0.49 | 83.41 | 4.49 | 3.77 | 73.24 | 1.97 | 1.29 |
| | Online LiRA | 72.42 | 3.90 | 3.01 | 91.52 | 13.23 | 3.83 | 82.23 | 6.78 | 3.51 |
| | Offline LiRA | 55.63 | 1.17 | 0.61 | 76.11 | 1.92 | 0.99 | 63.49 | 1.11 | 0.95 |
| | Online RMIA | 72.08 | 5.60 | 2.60 | 90.84 | 8.12 | 6.34 | 82.51 | 8.63 | 4.43 |
| | Offline RMIA | 71.50 | 5.35 | 3.61 | 90.62 | 9.54 | 7.96 | 82.17 | 7.16 | 5.43 |
| LiRA-based | Attack-R | 47.33 | 0.21 | 0.14 | 49.21 | 0.14 | 0.04 | 46.03 | 0.14 | 0.10 |
| | Online LiRA | 49.22 | 0.65 | 0.35 | 49.97 | 0.87 | 0.60 | 46.47 | 0.22 | 0.10 |
| | Offline LiRA | 52.57 | 0.67 | 0.35 | 50.22 | 0.76 | 0.30 | 50.25 | 0.25 | 0.13 |
| | Online RMIA | 48.13 | 0.41 | 0.0 | 50.55 | 0.00 | 0.00 | 47.39 | 0.00 | 0.00 |
| | Offline RMIA | 45.76 | 0.64 | 0.44 | 49.40 | 0.18 | 0.16 | 44.42 | 0.17 | 0.16 |
| RMIA-based | Attack-R | 46.75 | 0.32 | 0.02 | 48.23 | 0.00 | 0.00 | 47.59 | 0.00 | 0.00 |
| | Online LiRA | 51.76 | 0.83 | 0.55 | 52.33 | 0.60 | 0.32 | 49.28 | 0.24 | 0.14 |
| | Offline LiRA | 53.62 | 0.66 | 0.57 | 48.98 | 0.00 | 0.00 | 52.71 | 0.11 | 0.04 |
| | Online RMIA | 48.65 | 0.00 | 0.00 | 47.68 | 0.00 | 0.00 | 48.32 | 0.00 | 0.00 |
| | Offline RMIA | 46.39 | 0.93 | 0.30 | 49.16 | 0.10 | 0.04 | 46.52 | 0.14 | 0.00 |

**MIA Baselines.** We consider three SOTA MIAs as baselines: Attack-R (Ye et al., 2022), LiRA (online and offline)(Carlini et al., 2022), and RMIA (online and offline) (Zarifzadeh et al., 2024). For the reference models used by these baselines, we employ the same Wide-ResNetbackbone and training protocol, and resample 50/50 member versus nonmember splits consistent with the target-model setting. This design ensures that, for each query point, it appears in the training set of half of the reference models and is excluded from the training set of the other half. For each baseline, we apply the same set of data augmentations as in RMIA (Zarifzadeh et al., 2024) to enhance attack power.

**Evaluation metrics.** We evaluate forged examples along two aspects. First, we assess whether forged examples evade MIAs. We report standard MIA metrics: the area under the receiver operating characteristic curve (AUC score), and the true positive rate (TPR) at extremely low false positive rates (FPRs). Specifically, we evaluate at FPRs of 0.01% and 0.0%. We deem forgeability successful when MIAs cannot distinguish forged examples from genuine members. Second, to quantify the discrepancy between forged and unperturbed data, we compute the $\ell_\infty$ norm between each forged example and its counterpart and the average change in accuracy across all reference models when evaluated before and after perturbation. These measures capture the impact of the perturbation on data utility.

## 4.2 MAIN RESULTS

**Assessment of Indistinguishability** To comprehensively assess indistinguishability, we compare MIA performance across three cohorts: normal non-members, forged examples constructed based on LiRA signals, and forged examples constructed based on RMIA signals. The results in Tab. 1 show that when forged examples replace non-members in the evaluation set, MIA performance becomes insensitive to the *guiding signal*. Across attacks, the metrics drop to near random guessing, with an average AUC of 48.83%, regardless of whether the forged examples are constructed using LiRA or RMIA signals. For instance, when evaluating on normal non-members, strong online MIAs, specifically online RMIA and online LiRA, achieve a high attack success rate on CIFAR-100 with an average AUC of 91.18% and a TPR at 0.01% FPR of 10.68%. While evaluating on forged examples, the AUC consistently drops to approximately 50% and TPR decreases by a factor of 11.8, effectively neutralizing the adversary's advantage. The same pattern holds for offline MIAs. Using forged examples lowers AUC to chance and reduces TPR at extremely low FPRs to negligible levels.

Table 2: Effect of the number of data augmentations used by online LiRA on MIA performances against forged examples on CIFAR-10.

| Metric | # of Data Augmentations | | | | | | | |
|--------|------|------|------|------|------|------|------|------|
| | 2 | 4 | 6 | 8 | 10 | 14 | 16 | 18 |
| AUC | 50.10 | 50.08 | 50.14 | 50.17 | 49.97 | 49.35 | 49.23 | 49.22 |
| TPR@0.01% FPR | 0.44 | 0.47 | 0.40 | 0.42 | 0.51 | 0.65 | 0.67 | 0.65 |
| TPR@0.0% FPR | 0.20 | 0.22 | 0.28 | 0.28 | 0.40 | 0.44 | 0.39 | 0.35 |

Table 3: Effect of the number of reference models used by online LiRA on MIA performances against forged examples on CIFAR-10.

| Metric | # of Reference Models | | | | |
|--------|------|------|------|------|------|
| | 2 | 64 | 128 | 192 | 254 |
| AUC | 51.45 | 49.74 | 49.38 | 49.32 | 49.22 |
| TPR@0.01% FPR | 0.17 | 0.43 | 0.66 | 0.63 | 0.65 |
| TPR@0.0% FPR | 0.03 | 0.21 | 0.28 | 0.33 | 0.35 |

We ablate the number of data augmentations (Tab. 2) and the number of reference models (Tab. 3) to evaluate the indistinguishability of forged examples under different MIA configurations. As illustrated in Tab. 2, the AUC remains near 50%, and TPR at FPRs of 0.01% and 0% is always below 0.7% and 0.05%, respectively, regardless of the number of augmentations used by the MIA. Likewise, Tab. 3 shows a consistent near-50% AUC and negligible TPR at these low FPRs when varying the number of reference models from 2 to 254. Overall, the indistinguishability of forged examples is insensitive to the MIA configurations. Additional ablation studies demonstrating that forged examples can mislead MIAs even when compared against other non-members are in App. D.

**Assessment of Imperceptibility** We assess the imperceptibility from the aspects of input contents and data utility, which are measured by $\ell_\infty$-norm distance between the original and forged input, and the average accuracy of reference models when evaluated on forged examples, respectively. As shown in Tab. 4, the input difference is negligible, with $\ell_\infty$-norm close to zero (e.g., $\leq 0.007$). Moreover, the average accuracy of reference models on forged examples matches that on original non-members, which indicates that forged examples preserve utility.

Table 4: Per-dataset $\ell_\infty$-norm difference between original data and forged data, and corresponding accuracy compared to original data. For Accuracy, the value in parentheses indicates the difference relative to the accuracy on original data (%).

| PoF signal | CIFAR-10 | | CIFAR-100 | | CINIC-10 | |
|------------|-----------------|---------------|-----------------|---------------|-----------------|---------------|
| | $\ell_\infty$-norm | Accuracy | $\ell_\infty$-norm | Accuracy | $\ell_\infty$-norm | Accuracy |
| LiRA-based | 0.0012 | 98.94(+2.93) | 0.0035 | 90.45(+6.96) | 0.0025 | 94.20(+5.59) |
| RMIA-based | 0.0018 | 98.80(+2.79) | 0.0070 | 89.37(+5.88) | 0.0042 | 94.03(+5.42) |

## 5 CONCLUSION

This work introduces Proof-of-Forgeability (PoF) as a practical repudiation against membership inference claims. Instead of disclosing the training set or producing time-consuming per-claim Proof-of-Repudiation logs, PoF shows that the MIA-based claims are forgeable by constructing forged examples that MIAs would misclassify as members. This undermines the validity of MIA and enables an universal repudiation against all membership inference claims. We present a systematic procedure for forging from non-member queries via imperceptible noise. For each query, MLSE infers the member-like signal. We then determine the perturbation magnitude using a first-order Taylor expansion and adjust the corresponding adjustment to the input. The resulting forged examples consistently fool MIAs while remaining imperceptibly close to the original non-members.

# 6 ETHICS STATEMENT

Our study investigates the generation of forged examples to evade MIAs and hence discredits membership inference claims produced by these attacks. These findings motivate a reassessment of how current MIAs quantify privacy leakage and call for robust MIAs that remain effective against forged examples.

We acknowledge that PoF could be harmful if misused. To mitigate this risk, our work is framed as an academic study of reliability in membership inference attacks and emphasizes responsible communication of findings. The analysis and results are intended solely for scientific research, with an emphasis on transparency and reproducibility.

# 7 REPRODUCIBILITY STATEMENT

We provide the pseudo-code for generating forged examples in App. A and describe experimental setups in §4.1 and App. C. To support reproducibility, we release our implementation in the following anonymized repository: https://anonymous.4open.science/r/348079E324098F428C/

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

# A PSEUDO-CODE FOR FORGED DATA GENERATION

In this section, We present the pseudo-code of forged data generation used in PoF. This algorithm bridges signal space and data space, ensuring that non-member examples are perturbed just enough to align their attack signals with the member distribution. Alg. 1 outlines the pseudo-code. The input to this algorithm includes the target model $f_\theta$, a target data point $(\mathbf{x}, \mathbf{y})$, member and non-member datasets $(D_{\text{mem}}, D_{\text{non}})$, the iteration budget $T$, a perturbation bound $\epsilon$, and a scalar signal function $J(\cdot)$ (e.g., true-label confidence). We first compute attack signals on both $D_{\text{mem}}$ and $D_{\text{non}}$ under $f_\theta$ and construct a MLSE $s^F = F_M^{-1}(F_N(s))$ to obtain the member-like target $s^F$. To map the desired signal shift to the input domain, we approximate the perturbation scale via a first-order Taylor expansion, and perform projected sign of gradient updates within the $\ell_\infty$ ball of radius $\epsilon$. The procedure stops when the forged signal reaches $s^F$ or when $T$ steps are exhausted.

---

**Algorithm 1** Forged Data Generation

---

**Require:** Target data point $(\mathbf{x}, \mathbf{y})$; target model $f_\theta$; member set $D_{\text{mem}}$, non-member set $D_{\text{non}}$; iterations $T$; norm bound $\epsilon$; signal function $J(\cdot)$; method $D(\cdot)$ producing $m$ augmented samples; quantile functions $F(\cdot)$

**Ensure:** Forged example $\hat{\mathbf{x}}$

1: **Compute signal distribution on the target model:**
$$S^M = \{ J(f_\theta(\mathbf{x}^M), \mathbf{y}^M) \mid (\mathbf{x}^M, \mathbf{y}^M) \in D_{\text{mem}} \}$$
$$S^N = \{ J(f_\theta(\mathbf{x}^N), \mathbf{y}^N) \mid (\mathbf{x}^N, \mathbf{y}^N) \in D_{\text{non}} \}$$

2: $s^F = F_M^{-1}(F_N(s))$            ▷ quantile matching for Eq. 4

3: $\hat{\mathbf{x}}_0 = \mathbf{x}, \quad s_0 = J(f_\theta(\hat{\mathbf{x}}_0), \mathbf{y})$

4: $\eta = \dfrac{s_0 - s^F}{\|\nabla_{\mathbf{x}} J(f_\theta(\hat{\mathbf{x}}_0), \mathbf{y})\|_1}$

5: **for** $t = 0$ to $T - 1$ **do**

6:      $g_t = \frac{1}{m} \sum_{i=0}^{m-1} \nabla_{\mathbf{x}} J(f_\theta(D(\hat{\mathbf{x}}_t)_i), \mathbf{y})$

7:      $\hat{\mathbf{x}}_{t+1} = \text{Proj}_x^\epsilon(\hat{\mathbf{x}}_t + \eta \cdot \text{sign}(g_t))$

8:      $s_{t+1} = J(f_\theta(\hat{\mathbf{x}}_{t+1}), \mathbf{y})$

9:      **if** $s_{t+1} > s^F$ **then**

10:          **break**

11:      **end if**

12: **end for**

13: **return** $\hat{\mathbf{x}} = \hat{\mathbf{x}}_{t+1}$

---

# B PROOF DETAILS

## B.1 PROOF OF LEMMA 3.3

*Proof.* The first-order Taylor expansion of $s$ around $\mathbf{x}$ for a small perturbation $\delta_{\mathbf{x}}$ is:

$$s(\mathbf{x} + \delta_{\mathbf{x}}) \approx s(\mathbf{x}) + \mathbf{g}^T \delta_{\mathbf{x}} + \mathcal{O}(\|\delta_{\mathbf{x}}\|^2), \tag{9}$$

where the higher-order terms can be neglected for a small scale of $\epsilon$. Therefore, for the target signal change of $\Delta s = s(\mathbf{x} + \delta_{\mathbf{x}}) - s(\mathbf{x})$, we have:

$$\Delta s \approx \mathbf{g}^T \delta_{\mathbf{x}}. \tag{10}$$

Choosing the direction $\delta_{\mathbf{x}} = \epsilon \, \text{sign}(\mathbf{g})$ aligns with the fast gradient sign method for efficient perturbation, maximizing the change under $L_1$ constraints:

$$\mathbf{g}^\top \delta_{\mathbf{x}} = \mathbf{g}^\top (\epsilon \, \text{sign}(\mathbf{g})) = \epsilon \left( |\mathbf{g}|^\top \cdot \mathbf{1} \right) = \epsilon \|\mathbf{g}\|_1, \tag{11}$$

since $\mathbf{g}_i \times \text{sign}(\mathbf{g}_i) = |\mathbf{g}_i|$. Solving for $\epsilon$:

$$\epsilon = \frac{\Delta s}{\|\mathbf{g}\|_1}. \tag{12}$$

$\square$

## B.2 PROOF OF LEMMA 3.1

*Proof.* We prove this in three steps: (1) pushforward, (2) Monotonicity and $\mu$-almost everywhere uniqueness, and (3) optimality among all measure-preserving maps via cyclical monotonicity and the rearrangement inequality.

**Step 1**. Let $U \sim \text{Uniform}[0, 1]$. Then $Q_\mu(U) \sim \mu$ and $Q_\nu(U) \sim \nu$. By construction,

$$T\big(Q_\mu(U)\big) = Q_\nu\big(F_\mu(Q_\mu(U))\big) = Q_\nu(U), \tag{13}$$

so $T(Q_\mu(U)) \sim \nu$. Hence $T$ pushes $\mu$ forward to $\nu$, denoted by $T_\#\mu = \nu$.

**Step 2**. Both $F_\mu$ and $Q_\nu$ are non-decreasing; hence $T = Q_\nu \circ F_\mu$ is also non-decreasing.

Let $T' : \mathbb{R} \to \mathbb{R}$ be any non-decreasing map with $T'_\#\mu = \nu$. Fix a continuity point $u \in (0, 1)$ of $Q_\nu$ and set $s_u := Q_\mu(u)$. Then

$$\nu\big((-\infty, T'(s_u)]\big) = \mu\big(\{s : T'(s) \le T'(s_u)\}\big) \quad \ge \quad \mu\big((-\infty, s_u]\big) = u, \tag{14}$$

so $T'(s_u) \ge Q_\nu(u)$. Repeating with $u' < u$ and using monotonicity gives $T'(s_u) \le Q_\nu(u)$. Hence $T'(s_u) = Q_\nu(u)$. Since the continuity points of $Q_\nu$ have full Lebesgue measure in $(0, 1)$ and $u = F_\mu(s)$ holds for $\mu$-a.e. $s$, we conclude $T'(s) = Q_\nu(F_\mu(s)) = T(s)$ for $\mu$-a.e. $s$.

**Step 3**. The fundamental theorem of optimal transport (see Theorem 1.13 of Ambrosio et al. (2005)) claims that a transport plan $\gamma$ is optimal if and only if its support is $c$-cyclically monotone, where $c(s_1, s_2) = h(s_2 - s_1)$. A set $\Gamma \subseteq \mathbb{R} \times \mathbb{R}$ is $c$-cyclically monotone if, for any finite sequence $\{(s_i, t_i)\}_{i=1}^N \subseteq \Gamma$, the inequality

$$\sum_{i=1}^N c(s_i, t_i) \le \sum_{i=1}^N c(s_i, t_{\sigma(i)}), \tag{15}$$

holds for all permutations $\sigma$ of $\{1, \ldots, N\}$.

For $N = 2$, this reduces to $c(s_1, t_1) + c(s_2, t_2) \le c(s_1, t_2) + c(s_2, t_1)$. Assume by contradiction that an optimal plan has a crossing in its support: $s_1 < s_2$ but $t_1 > t_2$. Let $\delta = s_2 - s_1 > 0$, $a = t_1 - s_1$, $b = t_2 - s_2$. The inequality becomes $h(a) + h(b) \le h(a + \delta) + h(b - \delta)$. Set $k = \delta/(a - b)$ (assuming $a > b$ for the crossing; otherwise swap). If $k \in (0, 1)$, strict convexity of $h$ implies $h(a + \delta) + h(b - \delta) < h(a) + h(b)$, contradicting the $\le$. Thus, optimal supports must be graph-monotone (non-crossing).

Given strictly increasing CDFs, the measures have no atoms and are continuous, so the monotone transport plan is unique and induced by $T$. Since an optimal plan exists and must be monotone, it coincides with the plan from $T$, making $T$ optimal. Complementarily, the rearrangement inequality asserts that for non-decreasing sequences, the minimal cost for convex $h$ is achieved by sorted (monotone) pairings, equivalent to quantile matching (see Theorem 2.12 of Villani et al. (2008)). For quadratic cost, $T$ explicitly minimizes the Wasserstein-2 distance.

$\square$

## B.3 PROOF OF THOREM 3.2

*Proof.* This derivation relies on the properties of the normal distribution and quantile matching, which aligns the CDFs of two distributions to make them identical.

Note that the CDF of a normal distribution $\mathcal{N}(\mu, \sigma^2)$ is $F(s) = \Phi\left(\frac{s-\mu}{\sigma}\right)$, where $\Phi$ is the CDF of the standard normal distribution $\mathcal{N}(0, 1)$. The quantile function (i.e., inverse CDF) for $\mathcal{N}(\mu, \sigma^2)$ is

$$F^{-1}(u) = \mu + \sigma\,\Phi^{-1}(u), \quad u \in (0, 1). \tag{16}$$

Suppose that $F_M, F_N$ are CDFs for the $S_M \sim \mathcal{N}(\mu_M, \sigma_M^2), S_N \sim \mathcal{N}(\mu_N, \sigma_N^2)$. The quantile matching defines the transformation by $T(\cdot) = F_M^{-1}(F_N(\cdot))$, which pushes the non-member distribution $S_N$ to the member distribution $S_M$, ensuring $T(S_N) := S_F \overset{d}{\sim} S_M$. Substituting the

expressions gives:

$$F_N(s_i^N) = \Phi\left(\frac{s_i^N - \mu_N}{\sigma_N}\right), \tag{17}$$

$$T(s_i^N) = F_M^{-1}\left(\Phi\left(\frac{s_i^N - \mu_N}{\sigma_N}\right)\right) = \mu_M + \sigma_M \Phi^{-1}\left(\Phi\left(\frac{s_i^N - \mu_N}{\sigma_N}\right)\right). \tag{18}$$

Since $\Phi^{-1} \circ \Phi(z) = z$ for $z \in \mathbb{R}$, we have $\Phi^{-1}\left(\Phi\left(\frac{s_i^N - \mu_N}{\sigma_N}\right)\right) = \frac{s_i^N - \mu_N}{\sigma_N}$. Thus, Equations 18 simplifies to

$$T(s_i^N) := s_i^F = \mu_M + \sigma_M \cdot \frac{s_i^N - \mu_N}{\sigma_N} = \mu_M + \frac{\sigma_M}{\sigma_N}(s_i^N - \mu_N). \tag{19}$$

This is an affine transformation, which preserves the Gaussian nature of the distribution $S_F$.

The target signal change is

$$\Delta s_i = s_i^F - s_i^N = \left(\mu_M + \frac{\sigma_M}{\sigma_N}(s_i^N - \mu_N)\right) - s_i^N = \mu_M - \mu_N + \frac{\sigma_M}{\sigma_N}(s_i^N - \mu_N) - (s_i^N - \mu_N). \tag{20}$$

Rearranging terms:

$$\Delta s_i = (\mu_M - \mu_N) + \left(\frac{\sigma_M}{\sigma_N} - 1\right)(s_i^N - \mu_N). \tag{21}$$

$\square$

## C    DETAILED EXPERIMENTAL SETUP

In this subsection, we provide a detailed introduction of experimental settings.

### C.1    MIA SETUP

for each MIA method, we use their default hyperparameters and the implementation from RMIA repository. For RMIA, the $\gamma$ is set to 1 for CIFAR-100 and 2 for other dataset. The soft-margin $m$ and the order $n$ in Taylor-based functions are 0.6 and 4, respectively accorss all datasets.

### C.2    QUANTILE MATCHING SETUP

ML models trained on the similar data usually share similar decision boundaries in input space. These boundaries are locally sensitive , such that even the small can move inputs across them for multiple models simultaneously,enabling the well-known phenomenon of adversarial transferability (Szegedy et al., 2013). However, the transferability is inherently imperfect since models rarely share identical decision boundaries due to the model specificity (e.g., initialization, architecture). As a result, exact quantile matching performed on the target model may not perfectly align with the distributions observed on reference models. To address this mismatch, we introduce an *excessive ratio* $\kappa$, which slightly increases the mapped quantile position of each non-member signal. Intuitively, $\kappa$ controls a small upward shift in the quantile mapping, ensuring that forged signals are pushed slightly further toward the member distribution, thereby improving robustness to slightly misaligned decision boundaries between the target model and reference models. Formally, for a non-member signal $s_i^N$, the forged member-like signal transformed from Eq. 4 into the refined formulation:

$$s_i^F = F_M^{-1}\left(\min\{F_N(s_i^N) + \kappa, 1\}\right). \tag{22}$$

The clipping at 1 ensures that the shifted quantile remains valid. The values of $\kappa$ used in our experiments are summarized in Tab. 5.

### C.3    GUIDING SIGNALS DESCRIPTION

Our proposed PoF framework generates forged examples by explicitly aligning attack signals used in SOTA MIAs. In particular, we instantiate it primarily using both *LiRA-based* and *LiRA-based* signals. Other type of signals may provide even stronger performance for generating forged examples. we leave signal exploration for future work.

Table 5: optimal excessive ratio $\kappa$ for each dataset and each aligned signal.

| Dataset | LiRA-based Signal | RMIA-based Signal |
|---|---|---|
| CIFAR-10 | 1.5% | 3% |
| CIFAR-100 | 0.75% | 3% |
| CINIC-10 | 0.75% | 1.5% |

Table 6: Results on CIFAR-10 under different scenarios. We report AUC and TPR at two extremely low FPR levels (0.01% and 0.0%).

| Scenario | Attack | AUC | TPR@FPR | |
|---|---|---|---|---|
| | | | 0.01% | 0.0% |
| Original non-members v.s. Other non-members | Online LiRA | 50.55 | 0.06 | 0.06 |
| | Online RMIA | 50.22 | 0.01 | 0.0 |
| Forged examples v.s. Other non-members | Online LiRA | 73.38 | 1.28 | 1.09 |
| | Online RMIA | 73.47 | 0.60 | 0.14 |

**LiRA-based Guiding Signal.**  Following Carlini et al. (2022), we adopt TLC as the guiding signal. For each query $(x, y)$, the TLC is defined as the model $f_{\theta_t}$'s logit for the ground-truth class, $p = f_{\theta_t(x)_y}$. We further apply logit scaling, $\phi(p) = \log \frac{p}{1-p}$. Hence, guiding signal function becomes $s(f_{\theta_t}(x), y) = \phi(p)$.

**RMIA-based Guiding Signal.**  We also adopt the Soft-Margin-Taylor-Softmax signal (Banerjee et al., 2021) used in RMIA as the guiding signal. Specifically, let $g_i$ denote the logit for class $i$ and $T$ controls temperature. Define $c_i = g_i/T$ and the $n$-th order Taylor approximation of the exponential by

$$\text{apx}(a) = \sum_{k=0}^{n} \frac{a^k}{k!}. \tag{23}$$

With a soft-margin hyperparameter $m \geq 0$ applied to the true class only, the guiding signal function for a sample $(x, y)$ under model $\theta_t$ is

$$s(f_{\theta_t}(x), y) = \frac{\text{apx}(c_y - m)}{\text{apx}(c_y - m) + \sum_{i \neq y} \text{apx}(c_i)}. \tag{24}$$

## D  ADDITIONAL RESULTS

We examine whether forged examples can be distinguished from other non-members. Here, the forged examples are generated by adding adversarial noise to the original non-members. This design enables a direct comparison between non-members and forged examples, highlighting how forged examples can still be misclassified as members by MIAs, though they are not part of training. As shown in Tab. 6, when comparing original non-members against other non-members, MIAs perform no better than random guessing (e.g., AUC $\approx 50\%$). Conversely, when forged examples are compared against other non-members, both Online LiRA and Online RMIA demonstrate substantially higher AUC and TPR at extremely low FPRs, indicating that MIAs misclassify forged examples as members, thereby validating the indistinguishability induced by PoF.

## E  VISUALIZATION OF FORGED EXAMPLES

We present several representative visualizations of the forged examples in Fig. 3.

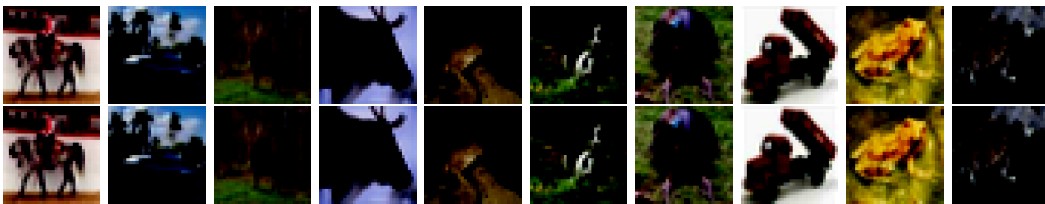

Figure 3: Comparison between forged examples (top row) and their corresponding benign samples (bottom row) on CIFAR10 using LiRA-based Signal.

## F  LLM USAGE DISCLOSURE

We only used LLMs as a writing assistant to polish the language of the manuscript. The LLM was used only for stylistic refinement and improving readability. It did not contribute to research ideation, experimental design, or interpretation of results. All conceptual contributions, methodology, and experiments were designed and conducted entirely by the authors.

