# OpenReview forum: "Proof of Forgeability: Universal Repudiation against Membership Inference Attacks"
_ICLR.cc/2026/Conference — Submitted to ICLR 2026_

### Official Review · Reviewer_2uSD · 2025-10-30

**Soundness:** 2
**Presentation:** 2
**Contribution:** 2
**Rating:** 2
**Confidence:** 4

**Summary:**

The paper studies the problem of refuting membership inference claims, where the goal is: given a trained model, how can one craft a data record, on which a specific membership inference attack's membership inference claim is wrong.  They proposes a method that adds minimal imperceptible perturbations to non-member samples so as to produce member-like signals. To solve this problem, they perform quantile matching to identify the member score value that matches the rank of the non-member score in non-member distribution, and then prove that quantile matching is one-dimensional optimal transport in score space. Finally, they use a Taylor expansion based technique to compute the approximate direction and magnitude of input perturbation needed to move non-member signal to its corresponding member signal. Experiments show the proposed method successful forge examples that are non-member, yet state-of-the-art MIA methods fail to correctly infer as non-member.

**Strengths:**

The problem of refuting membership inference has significant implication for data usage inference and copyright, and would boost the understanding of precise meaning and limitations of membership inference attack.

**Weaknesses:**

1.  The paper deem forged examples as non-member simply by input similarity to a ground-truth non-member. But this definition is flawed as membership inference targets member/non-member defined by strict game formulation and fresh randomness, **prior to releasing the trained model**. In this paper, the data records are forged after observing the trained model, making it arguable to deem them as non-member. A more convincing refutation, is to perturb data a priori to training, and then train model including/excluding the perturbed record, and show that the MIA accuracy drops to near random guess.
2. The authors only study one direction of refuting membership inference, i.e., perturbing non-member data to make it member-like. This is arguably the less interesting case, compared to perturbing a member to make it more non-member like. Indeed, in most copyright or data usage inference problems, membership inference is used for detecting usage, rather than for arguing one did not use certain data. The authors should at least clarify why they focus on this particular direction, and how the insights could generalize to the other direction.

**Questions:**

See weaknesses.

---

> ### Author Response · Authors · 2025-11-29
> **Response to Reviewer 2uSD (Part 1)**
>
> Thank you for the detailed review and valuable questions. For your convenience, we first clarify our threat model as below.
>
> ## Threat Model
>
> In our setting, we assume two main parties and a mediated arbiter. The plaintiff (the data curator) uses an MIA to claim that some of their data points (the queried points) were used to train the defendant’s model without authorization, while the defendant (the model owner) disputes this claim, and the arbiter evaluates the evidence from both sides. Plaintiff makes a membership claim on the queried points by 1) collecting a set of purportedly representative non-members, 2) applying a SOTA MIA to compute the attack signals for queried points and non‑members, and 3) choosing a decision threshold to balance true‑positive and false‑positive rates between the queried points (treated as members) and these non‑members.
>
> This defines the plaintiff’s threat model: they have black‑box access to the target model, knowledge of their own queried data, and the ability to run an MIA. **Their claim implicitly assumes that the chosen non‑members are representative**, so large deviations between the MIA scores of queried points and those of non‑members are taken as evidence of membership. However, the defendant can argue that this representativeness assumption is **unjustified**: the plaintiff’s non‑members may have been chosen inadvertently to support the claim. The defendant can instead provide an alternative set of non‑members (forged examples) that flips the MIA outcome, thereby convincing the arbiter that the plaintiff’s membership claim is unreliable. Concretely, the defendant uses the same queried  points presented by the plaintiff, but pairs them with forged examples to show that the MIA produces a different membership conclusion. This is not to decide membership directly (which the defendant may already know), but to show the arbiter that MIA‑supported evidence itself is unreliable as legal proof. We term the process as **proof of forgeability (PoF)**.
>
> ## [W1] On the definition of forged examples as non-members
>
> We **do not** define non‑membership by “input similarity.” **In our setting, the defendant is the model owner and exactly knows which records were non‑members of the fixed training set.** PoF starts from such *true* non‑members and perturbs them into forged examples; by definition they remain non‑members for the target model.
>
> What PoF challenges is **not** the formal MIA game, but the **evidentiary use** of MIAs by the plaintiff: their claim relies on a **chosen** non‑member set and the resulting TPR–FPR trade‑off. **By presenting an alternative non‑member set (forged examples) on which the same MIA loses its TPR–FPR trade-off, the defendant undermines the assumed representativeness of the plaintiff’s non‑members.** The suggested “perturb a priori and retrain” design is complementary setup, but it targets a different, pre‑training scenario than our post‑hoc forensic setting.

---

> ### Author Response · Authors · 2025-11-29
> **Response to Reviewer 2uSD (Part 2)**
>
> ## [W2] Why we focus on non-member → member-like
>
> In our **defensive, arbiter‑mediated** setting, the plaintiff uses MIAs to accuse the defendant of having trained on queried points. The defendant’s goal is to **challenge the strength of this claim**, not to detect unseen usage. In this context, the relevant is **non‑member → member‑like**: starting from known non‑members under the defendant’s control and showing that many of them can be perturbed to obtain member‑like MIA scores, **which in turn inflates the effective FPR and undermines the claimed ROC trade‑off.**
>
> We agree that the opposite direction (member → non‑member‑like) matters for usage‑detection scenarios (e.g., copyright claims). **Extending our signal‑space analysis to that setting is a natural next step**, and we will clarify this scope in the revision.

---

### Official Review · Reviewer_dZ9A · 2025-10-31

**Soundness:** 3
**Presentation:** 3
**Contribution:** 2
**Rating:** 4
**Confidence:** 4

**Summary:**

The paper introduces Proof of Forgeability (PoF), a defense mechanism against Membership Inference Attacks. Instead of refuting individual membership claims through computationally heavy Proof-of-Repudiation (PoR) logs, PoF proposes a universal repudiation approach: generating forged non-member examples that are misclassified by MIAs as members.
The key idea is to construct imperceptible perturbations on non-member samples using a Member-Like Signal Estimator (MLSE) derived via quantile matching and Taylor expansion, aligning their attack-signal distributions with members. Empirical results show that forged examples fool LiRA and RMIA attacks.

**Strengths:**

- The idea of leveraging adversarial perturbations as a means of repudiating membership inference is interesting.

- The theoretical framework that integrates quantile matching, optimal transport, and first-order Taylor expansion is well-formulated and mathematically solid to me.

**Weaknesses:**

- The evaluation focuses solely on MIAs that depend on distribution differences of output signals (e.g., LiRA, RMIA). It neglects other major categories of attacks, for example, reference-calibration-based methods (e.g., [R1]) and label-only MIAs (e.g., [R2]), which do not rely on confidence scores or logit access. Since PoF’s construction explicitly assumes access to continuous attack signals, its applicability to label-only or black-box MIAs is unclear.

- The empirical section lacks crucial implementation details needed for reproducibility, such as: (1) How many non-member samples are used to generate forged examples per dataset and model; (2) The sensitivity of results to the perturbation bound \epsilon.

- The main idea—perturbing non-member inputs until they appear as members—is ethically questionable, especially since the paper explicitly uses a courtroom analogy (Figure 1).
Such a mechanism effectively creates falsified evidence rather than verifiable proof, which undermines the integrity of the proposed legal scenario.

[R1] Watson, Lauren, et al. "On the Importance of Difficulty Calibration in Membership Inference Attacks." International Conference on Learning Representations. 2022.

[R2] Peng, Yuefeng, et al. "Oslo: one-shot label-only membership inference attacks." Advances in Neural Information Processing Systems 37 (2024): 62310-62333.

**Questions:**

- How would PoF perform against other major types of membership inference attacks, such as reference-calibration-based methods (e.g., [R1]) or label-only MIAs (e.g., [R2]), which do not rely on confidence scores or logit access?

- How many non-member samples are used to generate forged examples for each dataset and model?

- Does the approach risk producing falsified or misleading evidence rather than verifiable proof, thereby undermining the integrity of the intended legal scenario?

**Details Of Ethics Concerns:**

- The main idea—perturbing non-member inputs until they appear as members—is ethically questionable, especially since the paper explicitly uses a courtroom analogy (Figure 1).
Such a mechanism effectively creates falsified evidence rather than verifiable proof, which undermines the integrity of the proposed legal scenario.

---

> ### Author Response · Authors · 2025-11-29
> **Response to Reviewer dZ9A (Part 1)**
>
> ## [W3,Q3] Ethics concern
>
> We appreciate this concern and clarify both the intent and implications of our Proof of Forgability (PoF) framework.
>
> PoF has the same **defensive** goal as Proof of Repudiation (PoR): to allow the defendant to **challenge the plaintiff’s MIA‑supported claims, not to assert new claims about training data**. Our mechanism is explicitly about undermining the reliability of MIA‑based evidence, not about “creating falsified evidence” against membership.
>
> PoF exposes a **systematic vulnerability** of MIAs to the choice of the non‑member set: many genuine non‑members can be imperceptibly perturbed into forged examples that receive member‑like MIA scores, so an apparently strong TPR–FPR trade‑off can be manipulated. The courtroom analogy in Fig. 1 is intended to **highlight this risk** that MIA‑based claims can be manipulated and should not be treated as legal proof.
>
> As noted in the limitations, our findings **motivate a reassessment of how MIAs quantify privacy leakage and the design of more robust MIAs**, rather than encouraging misuse in real legal settings.
>
> We will add a clear statement that PoF is intended as a **diagnostic test** of MIA reliability, not as a tool for “producing falsified or misleading evidence”.

---

> ### Author Response · Authors · 2025-11-29
> **Response to Reviewer dZ9A (Part 2)**
>
> ## [Q2] How many non-member samples are used
>
> We thank the reviewer for pointing this out.
>
> In principle, PoF provides a construction pipeline that can be applied to **any number of non‑members**: each non‑member can be transformed into a forged example independently using the same procedure. **In our experiments, for each dataset and model, we apply this pipeline to the full non‑member set used to evaluate the corresponding MIA, thereby generating a forged counterpart for each such non‑member.** This allows us to show the arbiter a systematic degradation of the MIA’s TPR–FPR trade‑off, rather than a few isolated examples.
>
> ## [Q1] Applicability to other types of MIAs.
>
> We appreciate the question. This work is the **first** to formalize forgeability as a systematic vulnerability of MIAs. Our goal in this initial paper is to establish **feasibility**, and therefore, we focus on the strongest **SOTA MIAs** (i.e., LiRA and RMIA), which currently dominate practical and academic evaluations. Showing that PoF can reliably fool these SoTA attacks already demonstrates the core viability of the approach.
>
> We will clarify in the revised version that (1) the current work focuses on establishing feasibility against the strongest reference‑based MIAs, and (2) extending PoF/MLSE to calibration‑based and label‑only MIAs is an important and promising direction for future work.

---

### Official Review · Reviewer_6hLQ · 2025-10-31

**Soundness:** 1
**Presentation:** 2
**Contribution:** 1
**Rating:** 0
**Confidence:** 4

**Summary:**

This paper introduces a new approach to repudiate membership claims using membership inference, by showing membership inference attacks would falsely predict perturbed non-members as members. While it is interesting to see a paper dedicated to adversarial samples for MIAs, it falls short as a justified tool for membership repudiation.

**Strengths:**

1. Although known in the community, this is one of the first papers that actually presents a complete pipeline that generates adversarial data for membership inference attacks.
2. The experiments cover all popular and the state-of-the-art MIAs.

**Weaknesses:**

1. The motivation and the argument are incorrect. Showing that MIA would make mistakes on crafted adversarial data does not mean all of their other membership predictions are wrong. It is not enough to repudiate membership claims by showing MIAs make mistakes on a crafted adversarial data point that is not related to the target query. After all, all (automated) systems make mistakes, especially when used out of the designed data domains. And all MIAs have a designed false positive rate when setting their decision thresholds.
2. The authors argued that forged data is also from the same data distribution, but this claim is not corroborated by citations and does not stand. Adversarial data distribution is known to be different from natural data distribution even it is perceptually identical for humans. This results in the proposed framework testing on OOD data, and thus much less useful as repudiation proof.
3. Some important details of the experiments are missing. For example, the size of the $\ell$-infinity ball is not specified in the membership inference experiment. The authors also did not explain how online attacks are done on forged data. Online models for forged data need to be trained with forged data as part of the training set. The authors did not mention training these models. If the authors used the online models for the unperturbed data, they are actually offline, not online.
4. The main algorithm is basically an existing adversarial data generation algorithm with another objective that is based on membership inference signals. This significantly limits the novelty of the approach.
5. The writing is unscientific in several parts of the paper, especially in the introduction. A few words and sentences are written in an unnecessarily convoluted way. For example, ”mistakes the secondary for the primary" and "We answer in the affirmative".

**Questions:**

1. As mentioned in the weakness, the membership repudiation framework is not valid. To repudiate that model $f$ did not train on $x$, you need to show evidence that $f$, which is the observation, can be obtained without using $x$. This is the method used in the prior works (e.g. Kong et al. 2022). Your proposed framework argues MIAs on $f$ will make mistakes on specifically crafted data $z$ that is unrelated to $x$. This does not directly lead to the conclusion that $f$ is not trained on $x$. Hence, the proposed framework needs a redesign. I would suggest going along the false positive rate of MIAs.
2. What is the threat model? It seems that you need white-box access to the target model, training and test datasets, and the MIA used in membership claims to obtain IN and OUT signals for every data and to perform adversarial perturbation. If so, you can directly check the membership of your data query without using any MIA.
3. Given that adversarial data distribution is not identical to natural data distribution, forged data are essentially OOD data and it is not surprising for MIAs to misclassify them. What is your justification of using misclassified forged data as repudiation?
4. How do you launch online LiRA and RMIA on forged data as there are no reference models trained on forged data?
5. RMIA's signal is the percentage of population data points dominated by $x$, which is not Gaussian. How does your method, which assumes Gaussian, apply to RMIA?
6. “logit-scaled TLC" -> do you mean rescaled logits?
7. In the problem formulation, why is the model's output domain K-1 dimensional when there are K output classes?

**Minor comment**:

1. Besides images, how does the forging mechanism work on tabular data where the data is mostly categorical or binary values?
2. The citation for RMIA should be the ICML version instead of the arxiv version.
3. Kong et al. 2023 should be Kong et al. 2022
4. L465 da ta -> data

---

> ### Author Response · Authors · 2025-11-28
> **Response to Reviewer 6hLQ (Part 1)**
>
> We thank the reviewer for the detailed comments and constructive suggestions.
>
> To ensure a smooth response, we first clarify our threat model and motivation, and then address the specific concerns point by point.
>
> ## [Q2] Threat Model
>
> In our setting, we assume two main parties and a mediated arbiter. The plaintiff (the data curator) uses an MIA to claim that some of their data points (the queried points) were used to train the defendant’s model without authorization, while the defendant (the model owner) disputes this claim, and the arbiter evaluates the evidence from both sides. Plaintiff makes a membership claim on the queried points by 1) collecting a set of purportedly representative non-members, 2) applying a SOTA MIA to compute the attack signals for queried points and non‑members, and 3) choosing a decision threshold to balance true‑positive and false‑positive rates between the queried points (treated as members) and these non‑members.
>
> This defines the plaintiff’s threat model: they have black‑box access to the target model, knowledge of their own queried data, and the ability to run an MIA. **Their claim implicitly assumes that the chosen non‑members are representative**, so large deviations between the MIA scores of queried points and those of non‑members are taken as evidence of membership. However, the defendant can argue that this representativeness assumption is **unjustified**: the plaintiff’s non‑members may have been chosen inadvertently to support the claim. The defendant can instead provide an alternative set of non‑members (forged examples) that flips the MIA outcome, thereby convincing the arbiter that the plaintiff’s membership claim is unreliable. Concretely, the defendant uses the same queried  points presented by the plaintiff, but pairs them with forged examples to show that the MIA produces a different membership conclusion. This is not to decide membership directly (which the defendant may already know), but to show the arbiter that MIA‑supported evidence itself is unreliable as legal proof. We term the process as **proof of forgeability (PoF)**.
>
> Regarding the specific questions raised in Q2:
>
> 1. **Access to the target model.** PoF is operated by the defendant, who naturally has white box access to their own model and training data. This reflects the realistic forensic setting rather than granting extra attack power.
> 2. **Why use MIA and PoF if the defendant knows the ground truth?** As you correctly point out, if no arbiter were involved, the defendant could “directly check” membership. However, in our setting the arbiter must evaluate the plaintiff’s MIA-supported claim without trusting the defendant’s word alone. Instead of improving the defendant’s ability to infer membership, PoF targets the evidentiary value of MIAs in disputes. We show that the systematical vulnerability of **MIA outcomes to the choice of the non‑member set**, so they are not reliable enough to serve as forensic proof, even when the defendant already knows the ground truth.
>
> We will revise the Introduction and Method sections to clearly state this two‑party, arbiter‑mediated threat model and to emphasize that PoF addresses the reliability of  MIA-supported membership evidence, rather than enabling additional membership inference.

---

> ### Author Response · Authors · 2025-11-28
> **Response to Reviewer 6hLQ (Part 2)**
>
> ## [Q1, W1] Motivation
>
> We thank the reviewer for raising this concern and clarify the intended evidentiary scope of PoF.
>
> We agree that showing MIAs make mistakes on crafted inputs does not imply that all of their other predictions are incorrect, and **PoF does not claim this**. Our goal is instead to **challenge the evidentiary strength** of MIA‑supported claims that rely on a choice of the non‑member set to determine the decision threshold under our threat model. Concretely:
>
> 1. Regarding “all systems have a designed FPR”: We agree that MIAs are tuned to achieve a target FPR on the *chosen* non‑member set. PoF explicitly targets this dependence on the chosen non‑member set, rather than the mere existence of some errors. When we evaluate using members as positives and forged examples as negatives, any threshold that maintains high TPR on members necessarily induces a similarly high FPR on forged examples, so the attractive “high TPR @ low FPR” operating point claimed on the original non‑member set no longer holds. This directly undermines the reliability of the reported ROC curve as evidence.
>
> 2. Scope of repudiation: We do not claim that PoF proves “the model was never trained on the query” in the same **constructive sense** as PoR, which explicitly exhibits an alternative model. **Instead, PoF provides universal repudiation in an evidentiary sense:** once the defendant demonstrates that many genuine non‑members can be transformed into forged examples with member‑like signals via imperceptible perturbations, **the MIA’s claimed TPR–FPR trade‑off on the plaintiff’s chosen non‑members is no longer a trustworthy basis.** This is precisely “going along the false positive rate of MIAs”: PoF shows that the effective FPR can be made arbitrarily high on plausible non‑members, thereby weakening the probative value of the MIA output for the queried points.
>
> We will revise the introduction and the discussion around Fig. 1 to more clearly articulate this evidentiary notion of repudiation and its distinction from PoR.

---

> ### Author Response · Authors · 2025-11-28
> **Response to Reviewer 6hLQ (Part 3)**
>
> ## [W2, Q3] OOD issue
>
> We thank the reviewer for raising this valuable point. In our threat model, PoF only requires that forged examples be plausible training data from **the arbiter’s perspective**, not that they lie exactly on an (unknown) true data manifold. This is the relevant criterion for repudiation, since the arbiter only observes inputs and model outputs. We support this with two observations: (1) Forged examples differ from original non‑members by at most $L_\infty \le 0.007$, making them visually indistinguishable from natural data; (2) Tab. 4 shows that model utility on forged examples is similar to that on members.
>
> Thus, from the arbiter’s viewpoint, forged examples are effectively indistinguishable from natural data, so treating them as generic “OOD noise” is inappropriate in this evidentiary context. In the revision, we will also report results with standard OOD detectors, further showing that forged examples are not flagged as atypical and are treated similarly to natural data.
>
>
> ## [W3, Q4] “Online” LiRA/RMIA
>
> We apologize for the confusion and clarify the usage of “online” LiRA/RMIA in our setting.
>
> In our threat model, “online” LiRA/RMIA denote the MIAs that the plaintiff runs on the target model, using their queried points and a chosen non‑member set. Under PoF, we keep the same target model and same MIA configuration, and only replace the plaintiff’s non‑members with forged examples to show that the MIA conclusion can be flipped. No reference/shadow models are trained on forged data in this process.
>
> We will revise the terminology to make this explicit, and add experiments where reference models are trained with forged examples in their training sets to confirm robustness in that stronger setting.
>
>
> ## [W3] Some important details of the experiments are missing.
>
> The hard bound of the infinity ball is 0.06275 (16/255) for all datasets, which is a standard choice in previous adversarial example work (e.g., [1]). In practice, after applying our quantile matching, the actual perturbation magnitudes are an order  smaller: the realized noise satisfies $L_\infty \le 0.007$ (see Tab. 4). We will include these details in the revised version.
>
> [1] Dong, Yinpeng, et al. "Boosting adversarial attacks with momentum." *Proceedings of the IEEE conference on computer vision and pattern recognition*. 2018.

---

> ### Author Response · Authors · 2025-11-28
> **Response to Reviewer 6hLQ (Part 4)**
>
> ## [W4] Novelty
>
> Thanks for this comment; we clarify the novelty more concisely below.
>
> **Conceptual novelty.** Our main contribution is a novel repudiation paradigm against MIAs. Prior Proof of Repudiation (PoR) approaches require retraining on alternative datasets excluding the queried points, which is costly and often impractical. Our PoF instead discredits MIA‑based evidence by exploiting systematical vulnerabilities of MIA to the choice of the non‑member set, without any retraining. To the best of our knowledge, this **repudiation‑via‑forgeability** perspective for MIA evidence is novel and directly targets forensic scenarios, not standard robustness benchmarks.
>
> **Technical novelty.** While we use adversarial perturbation machinery, our method is not a generic adversarial example generation: (1) We explicitly model and match the member/non‑member attack‑signal distributions, rather than just flipping the decision boundary; and (2) we link input and attack‑signal spaces via a Taylor expansion and derive a closed‑form of the required perturbation size to control the MIA signal. These distribution‑matching and analytical components are specific to membership inference and, to our knowledge, are not present in prior adversarial‑example methods, so the contribution goes beyond “an existing adversarial algorithm with another objective.”
>
> ## [Q5,6,7, Minor comments]
>
> Thanks for these questions; we address them below.
>
> 1. “logit-scaled TLC” just refers to the “logit scaling to the model’s confidence” as defined in the LiRA paper, i.e., $log(p/1-p)$, where $p$ is the TLC. LiRA empirically shows that this logit‑scaled TLC is approximately Gaussian‑distributed.
>
> 2. **Attack signal and MIA score:** In RMIA, the defined “**MIA Score**” used for decision is indeed the percentage of percentage of population points dominated by a query, which is not Gaussian. However, RMIA internally operates on a continuous per‑sample attack signal (their Eq. (7)), analogous to the logit-scaled TLC used in LiRA. Our Gaussian assumption is made on this **internal continuous attack signal**, not on the **final MIA score**.
>
> 3.  $\Delta^{K-1}$ denotes the probability simplex over $K$ classes: model outputs are vectors in $\mathbb{R}^K$ whose entries are non‑negative and sum to 1. This constraint implies only $K-1$ degrees of freedom, hence the notation $\Delta^{K-1}$.
>
> 4. [Tabular data] Our current experiments focus on image data. We agree that designing such discrete perturbation schemes is a promising direction, and we will consider this as future work.
>
> 5. [Writing and citation issues] We thank the reviewer for the detailed notes. We will correct it in the revised manuscript.

---

### Official Review · Reviewer_r3Y6 · 2025-11-01

**Soundness:** 3
**Presentation:** 4
**Contribution:** 3
**Rating:** 6
**Confidence:** 4

**Summary:**

This paper studies the problem of repudiation in membership inference, showing that non-member samples can be transformed into forged members that are misclassified as members, thereby undermining the reliability of membership inference conclusions. The proposed method adds adversarial perturbations to non-member samples so that their membership signals resemble those of true members. Experiments demonstrate the effectiveness of the approach in achieving successful repudiation and reducing the credibility of membership inference results.

**Strengths:**

1. The paper is well-written, clearly structured, and easy to follow.
2. The proposed method is simple yet effective and intuitive. By constructing non-member samples that are misclassified as members, the work introduces a novel and practically meaningful way to repudiate membership inference results.
3. The paper provides both theoretical analysis and empirical evidence. Experiments are conducted using state-of-the-art MIA methods and demonstrate strong effectiveness, with attack performance (e.g., AUC) approaching random guessing after applying the proposed method.

**Weaknesses:**

1. The proposed method appears to be optimized primarily for RMIA and LiRA. Although these are state-of-the-art MIAs, it is unclear how well the approach generalizes to other classes of attacks, such as simple loss-based methods that do not rely on a reference model.

2. The paper does not discuss label-only MIAs [1][2][3][4], which also achieve strong performance and rely on label-based signals rather than loss or confidence. Since both label-only MIAs and the proposed repudiation approach introduce adversarial perturbations to the original samples, it is unclear whether these perturbations may interfere with each other and whether the proposed technique would remain effective under such conditions.

Overall, the paper is well-written and technically sound, with few major weaknesses. To further strengthen it, the authors might consider:
- Adding visual examples comparing non-member and forged samples to better illustrate the imperceptibility of the perturbations.
- Including a brief case study or qualitative example showing how the method alters the MIA signal and affects the final inference conclusions.

[1] *Label-Only Membership Inference Attacks*, ICML 2021
[2] *Membership Leakage in Label-Only Exposures*, CCS 2021
[3] *You Only Query Once: An Efficient Label-Only Membership Inference Attack*, ICLR 2024
[4] *OSLO: One-Shot Label-Only Membership Inference Attacks*, NeurIPS 2024

**Questions:**

1. Would adversarial purification methods, such as DiffuPure [1], affect the effectiveness of the proposed repudiation approach?

[1] *Diffusion Models for Adversarial Purification*, ICML 2022

---

> ### Author Response · Authors · 2025-12-01
> **Response to Reviewer r3Y6**
>
> We thank the   reviewer for the  thoughtful and encouraging feedback on our work.
>
> ## [W1, W2] Applicability to other types of MIAs.
>
> **On focusing on LiRA and RMIA.** This work is, to our knowledge, the first to formalize forgeability as a systematic vulnerability of MIAs and to propose PoF as a repudiation paradigm. To establish feasibility, we focus on SOTA reference‑based MIAs (e.g., LiRA and RMIA), which currently dominate practical and academic evaluations. Demonstrating that PoF reliably evades these attacks already fulfils the core objective of showing that MIA‑based evidence can be systematically undermined under our threat model.
>
> **On label‑only MIAs.** We agree that extending PoF beyond LiRA/RMIA is an important direction. Label‑only attacks rely on decision‑boundary geometry rather than continuous confidence signals, and are therefore fundamentally different from the reference‑based MIAs we target. Analyzing how PoF‑style forged examples interact with label‑only attack perturbations, and whether PoF can be adapted directly to their signal space, requires dedicated analysis, which we view as promising future work. We will add a discussion of this limitation and future direction in the revision.
>
> ## [Suggestion] visualization and qualitative examples
>
> We thank the reviewer for this helpful suggestion. We have added a visualization panel comparing non‑member and forged samples to illustrate the imperceptibility of the perturbations in App.E, and we will further include a brief qualitative example showing how the forged samples shift the MIA scores and flip the MIA outcomes to better convey the practical effect of PoF.
>
> ## [Q1] Effect of adversarial purification
>
> We thank the reviewer for raising this interesting question.
>
> Adversarial purification methods like DiffuPure act as an additional pre‑processing defense on the model inputs, rather than as part of the **MIA mechanism itself**. Analyzing how such purification interacts with PoF is indeed meaningful, but falls outside the scope of our initial study, which aims to establish the feasibility of forgeability as a systematic vulnerability of MIAs. We will add a discussion of this and highlight it as **promising future work** in the revised version.

---

### Meta-Review · Area_Chair_Pheq · 2025-12-09

**Summary:**

The main concerns of the reviewers were:
* Unclear description of the scenario and implications of the result
* Concerns on whether distribution of forged examples really matches target data
* Lack of technical details in presentation
* Focus on limited range of MIAs (mainly LiRA, RMIA)

**Reviewer Concerns:**

The authors respond to the reviewers concerns. In many cases, the responses promise revisions to the paper including adding new results, but no actual revision appears to have been made.

**Reviewer Scores:**

Given the failure of the authors to actually revise the paper and add the promised new results and explanations, I find it unlikely that all the negative reviewers would have changed their mind.

---

### Decision · Program_Chairs · 2026-01-26

Reject